# Does Distributed Leadership Deliver on Its Promises in Schools? Implications for Teachers' Work Satisfaction and Self-Efficacy

Mihai Tucaliuc [1,2], Petru Lucian Curșeu [1,3,*] and Arcadius Florin Muntean [1,2]

1 Department of Psychology, Faculty of Psychology and Educational Sciences, Babeș-Bolyai University, 400084 Cluj-Napoca, Romania; mihai.tucaliuc@ubbcluj.ro (M.T.); arcadius.muntean@ubbcluj.ro (A.F.M.)
2 Wellbeing Institute, 42254519 Cluj-Napoca, Romania
3 Department of Organization, Faculty of Management, Open Universiteit, 6419 Heerlen, The Netherlands
* Correspondence: petrucurseu@psychology.ro

**Abstract:** Although surrounded by theoretical confusion and methodological ambiguity, distributed leadership has been acclaimed as beneficial for teacher performance and student achievement. We set out to explore organizational identification and empowerment as two mechanisms that explain the positive and negative association between distributed leadership and teacher work-related outcomes. We build on social identity, social interdependence and cognitive schema theories to argue that teachers' cognitive dysfunctional schema of distrust and dependence moderate the association between distributed leadership on the one hand and organizational identification and empowerment on the other hand. We used multilevel mediation analyses to test our hypotheses in a sample of 3528 teachers, nested in 329 Romanian schools and our overall results reveal a negative association between distributed leadership and empowerment as well as organizational identification. Distrust cognitive schema accentuate the negative association between distributed leadership and empowerment, while dependence schema accentuate the negative association between distributed leadership and organizational identification. Finally, organizational identification mediates the association between distributed leadership and teachers' work self-efficacy as well as satisfaction, while empowerment only mediates the association between distributed leadership and work satisfaction.

**Keywords:** distributed leadership; teachers; work self-efficacy; work satisfaction; empowerment

## 1. Introduction

Distributed leadership is expected to bring many benefits for work performance and satisfaction because of increased employee participation in collaborative decision making, increased autonomy by sharing accountability among organizational members [1,2], increased commitment and staff empowerment [3–5]. Various definitions of distributed leadership co-exist [5–7], yet in line with a functional perspective [8,9] we define distributed leadership in schools as an allocation of various leadership tasks across multiple teachers that collaboratively fulfill the leadership roles and responsibilities. Building on effective principles of work design and organization, distributed leadership is expected to distribute managerial workload and to increase efficiency of work coordination [10]. Using a distributed cognition framework, the allocation of leadership tasks across different teachers is expected to reduce the cognitive load associated with leadership roles and responsibilities [1,5] and ultimately increase the organizational capacity of schools [11] by taking full advantage of their human and social capital [12].

Empirical evidence supports these claims showing that in schools, distributed leadership boosts teacher commitment and satisfaction, their academic capacity, and teaching effectiveness [9,13–15], facilitates learning in communities and contributes to improvement of academic capacity and learning effectiveness [11,16]. Moreover, recent studies show that



distributed leadership increases teacher trust and job satisfaction [9,17], fosters professional collaboration and teacher retention [13], and increases teachers' academic optimism, academic skills and expert knowledge [14,18,19]. Distributed leadership also increases student academic achievement [11]; therefore, in general, studies seem to depict a very optimistic image about handing out leadership functions to several teachers within schools.

A question still remains: What if distributed leadership is not a one size fits all practice in organizations and especially in the educational system? We build on some of the key tenets of the Social Identity Theory [20,21] to argue that distributed leadership could potentially have drawbacks and under certain circumstances dispersing leadership functions or roles in schools does not necessarily increase work performance and satisfaction. Individuals in general and teachers in particular identify with the social groups in which they work and the leaders of these groups are often prototypical representations of the group [21]. According to the core tenets of the Social Identity Theory [20,21], when leadership functions are distributed among different persons, the distinctiveness and the prototypicality of the leader is diminished. We have thus theoretical reasons to believe that distributed leadership in schools decreases group distinctiveness, creates fuzzy group boundaries and as a consequence reduces organizational identification. On the other hand, the core tenets of the Social Interdependence Theory [22] predict that distributed leadership functions can generate expectations of positive interdependence and ultimately increase empowerment. Therefore, distributed leadership may act like a double-edged sword in schools as it could reduce teachers' identification with the school and at the same time generate expectations of positive interdependence and empowerment. The literature to date did not comprehensively investigate competing mechanisms that could explain the beneficial or detrimental effects of distributed leadership in modern schools. We are not assuming here that distributed leadership is useful or not, but we argue that more research is needed in order to understand how, why and when distributed leadership may or may not work in schools. We build on Social Identity Theory [20,21] and on Social Interdependence Theory [22,23] to argue that organizational identification and teacher empowerment mediate the impact of distributed leadership on work performance and job satisfaction. Teachers filter the incoming information at work through their cognitive schema, namely the cognitive structures stored in their long-term memory and such personalized constructs [24–26] impact the way they look at distributed leadership too. Imagine a teacher that has a general tendency of not trusting others (distrust personal cognitive schema); for such a teacher, spreading leadership functions across various persons increases suspicion and decreases the sense of empowerment and identification with the school. Teachers scoring high on dependence (tend to seek consent from others in decision-making processes) also may dislike the allocation of leadership functions across different individuals in the school as they tend to seek approval from multiple colleagues while performing their tasks. We thus also introduce two individual level moderators in our analyses related to teachers' cognition, namely dysfunctional cognitive schema of dependence and distrust. We hypothesize that these two types of dysfunctional cognitive schema moderate the relationship between distributed leadership and teachers' sense of empowerment and identification with the school. Our paper presents one of the first empirical attempts that builds on relational arguments in order to disentangle the positive and detrimental influences of distributed leadership in schools. We hypothesize that organizational identification and empowerment explain the relationship between distributed leadership and teacher work self-efficacy and satisfaction with the school. Building on selective social perception, we include distrust and dependence cognitive schema as moderators in the relation between distributed leadership and teachers' empowerment and identification with the school.

## 2. Distributed Leadership, Organizational Identification and Empowerment

Organizational identification is an important part of one's social identity [20,21,27] and in line with the Optimal Distinctiveness extension [28] of the Social Identity Theory, one important mechanism that predicts the strength of identification with a particular social

group is its distinctiveness. Organizational distinctiveness refers to the employees' perception that their organization is special, unique and distinctive from other organizations [28]. The more distinctive an organization is, the stronger the tendency of its employees to identify with the organization as a unique and special organization can fulfill affiliation needs, as well as the need to be different and special [29,30]. An important source of organizational distinctiveness is its leaders; therefore, when prototypical leaders emphasize organizational distinctiveness and reinforce individuals' attachment to their organization, they create strong levels of organizational identification [21,27,31]. In line with the tenets of the Social Identity Theory and optimal distinctiveness, we expect that distributed leadership creates fuzzy relational boundaries, as it is unclear who is the prototypical leader expected to reinforce organizational distinctiveness. Leaders are prototypical images of the group [21,27,32]; therefore, multiple leaders may generate heterogeneous and incongruent images of organizational prototypes, blur group boundaries and as a consequence reduce the strength of organizational identification. We argue that under distributed leadership functions, employees may develop heterogeneous identification with the co-existing leaders and as a consequence diminish the strength of their organizational identification.

**Hypothesis 1 (H1):** *Distributed leadership has a negative association with organizational identification.*

Empowerment describes a social process in which actors (people, groups, organizations or communities) are given mastery over work-related aspects and have control over the tasks they are expected to perform [33]. In organizational settings, empowerment reflects efforts aimed at generating agency and proactivity among employees in order to increase task engagement and ultimately foster organizational effectiveness [34]. Most recent research considered empowerment as an integral part of distributed leadership practices [16,35]. Moreover, scales used to evaluate distributed leadership explicitly include dimensions like teacher empowerment, shared decision making and participation in decision-making processes concerning task planning and allocation [11,35,36]. When teachers are allowed participation in the decision-making processes aimed at influencing their work practices and tasks, they are likely to experience more positive interdependence (understand that in order to achieve their individual aims they need to support others to achieve their) and ultimately feel more empowered at work. In line with the Social Interdependence Theory [22,23], we argue that distributed leadership functions in schools increase the likelihood of experiencing empowerment.

**Hypothesis 2 (H2):** *Distributed leadership has a positive association with empowerment.*

### 3. Dysfunctional Cognitive Schema, Empowerment and Identification

Dysfunctional cognitive schemas are cognitive representations, often organized as complex beliefs systems that are derived from early life experience (including relations with relevant others) and shape the way in which external world is interpreted [25,37]. Such dysfunctional cognitions ultimately impact individual behavior during adulthood, as they incorporate conceptualizations of the self and relationships with others [26,37,38]. Early maladaptive schemas are developed in relation to childhood experiences, especially in relation to relevant others, and they comprise emotionally laden cognitions that once activated during adulthood may enact distorted interpretations of social stimuli and unhealthy reactions in interpersonal settings [38–40]. As cognitive structures interpose between stimuli and behaviors [25,37,40], we state that cognitive schemas are filters through which individuals organize incoming social information, including information stemming from social relationships and other relational work events. In particular, cognitive schema of distrust reflect the expectation that others will intentionally hurt, abuse, humiliate, cheat, lie, manipulate or take advantage during interpersonal interactions [39]. Once distrust schemas are activated, perceivers tend to filter and interpret incoming relational information with excessive scrutiny; they believe it to be suspicious and potentially harming.

Neutral interpersonal events at work may be treated as potentially harming by employees scoring high on distrust; therefore, we expect they will have a lower tendency to identify with the organization than employees scoring low on distrust. Moreover, distrust schemas may generate distorted interpretations of interactions with leaders and attribute malevolent management intentions of being tested, put under scrutiny or taken advantage of rather than empowering. As a consequence, we expect that employees scoring high on distrust tend to report less empowerment than employees scoring low on distrust.

**Hypothesis 3 (H3):** *Distrust has a negative association with organizational identification and empowerment.*

Dysfunctional cognitive schema of dependence reflects individuals' perception of being unable to handle daily responsibilities by themselves and feeling incompetent when asked to solve routine problems, making daily decisions or engaging in new tasks [26,39]. In educational settings, when dependence schema are activated, teachers' tendency of over-relying on their leaders and colleagues to perform their daily work-related tasks will increase, and they may feel less capable of doing work properly. We expect that teachers scoring high on dependence dysfunctional schema are less likely to feel empowered or capable of acting independently and performing their work-related tasks autonomously. In line with these arguments, we hypothesize:

**Hypothesis 4 (H4):** *Dependence has a negative association with empowerment.*

We believe that the relation between dependence and organizational identification is more complex and a two-folded argument leads to the formulation of competing hypotheses. In line with the need theories of motivation, employees strive to balance two simultaneous needs: the need for belonging and the need for being different [29]. The first one, belonging to the group, is reflected by a strong tendency to identify with the group and mobilize resources to emphasize group distinctiveness. The other one is the need to be different, reflected in higher role differentiation perceived within the group and emphasize individual rather than group distinctiveness [29]. The need to belong offers, therefore, a plausible argument for expecting a positive relationship between dependence cognitive schema and organizational identification. Given their lack of perceived personal autonomy, teachers scoring high on dependence cognitive schema will have a tendency to consider themselves "one with the organization", therefore display high levels of organizational identification.

**Hypothesis 5a (H5a):** *Dependence has a positive association with organizational identification.*

However, in line with some more arguments stemming from of Social Identity Theory, a key prerequisite for a strong social identification is the distinctiveness of the self-related cognitive schema and group-related cognitive schema [29,41]. In other words, in order to be able to identify oneself with the group, the cognitive schema related to self should be distinct and distinguishable from the group cognitive representation. Teachers scoring high on dependence dysfunctional schema have difficulties in defining themselves as autonomous in relation to their school. As such, when teachers score high on dependence schema, it is likely that the distinction between the self and the group is rather blurred. In the most extreme cases, one could imagine a situation in which the self-schema is diluted into the group schema. If the self-image is fully dependent on others' approval and actions, identification with a team or organization is difficult to conceptualize. In two experimental studies, Forehand, Deshpandé and Reed [42] showed that participants' social distinctiveness increased the salience of their social identity, such that socially distinctive individuals were more likely to identify themselves with their own social group. In line with these arguments, we could state that dependence-cognitive schema (low social distinctiveness) has a negative association with organizational identification. Given the two

opposing arguments, one derived from the need to belong and one derived from the need to be different [29] as well as the arguments related to the fluidity of the interplay between self-concept and social identity [43], we formulate a second competing hypothesis for the relation between dependence dysfunctional schema and organizational identification:

**Hypothesis 5b (H5b):** *Dependence has a negative association with organizational identification.*

## 4. The Interplay of Distributed Leadership and Dysfunctional Cognitive Schema

As stated above, we expect distributed leadership to generate fuzzy group boundaries and leader member relations. When followers join a new group, their relational identification with the prototypical leader can generalize to their identification with the group [32], so the more leaders and leadership functions in a group, the weaker organizational identification will be. Dysfunctional schema of distrust and dependence generate distorted interpretations of interpersonal and group boundaries that may accentuate the disruptive effects of distributed leadership on group boundaries and distinctiveness. In other words, we expect that in educational settings, it will be more difficult for a teacher scoring high on distrust to identify with the organization when the leadership functions are distributed. More leaders and distributed leadership responsibilities generate more ambiguity regarding roles, functions or tasks that lead to less clarity in terms of work expectations. We also expect that teachers scoring high on dependence cognitive schema could find it problematic to conceptualize their self-image when leadership is distributed because of the fuzzy relational boundaries.

**Hypothesis 6 (H6):** *Distrust (a) and dependence (b) accentuate the negative association between distributed leadership and organizational identification.*

We have already argued that multiple leaders can create more ambiguous group boundaries, reduce group distinctiveness and this ambiguity may increase when teachers have dysfunctional cognitive schemas. In educational settings, teachers scoring high on distrust cognitive schemas will be more skeptical regarding distributed leadership practices and may feel less empowered. Ambiguity regarding task, role or responsibilities are expected to further enforce distrust for these teachers, and they may think that the "actual reasons behind" may be other hidden objectives or the agendas of several leaders. For distrustful teachers, more people that fulfill leadership roles or functions in schools increase scrutiny and interpersonal suspicion; therefore, the level of trust regarding empowerment practices is more likely to decrease. To some extent, we could argue that teachers with dependence cognitive dysfunctional schema seek validation and support even when this is not needed; therefore, they will not feel empowered but rather burdened with the available autonomy. Multiple leaders and leadership functions distributed means more people to depend on and more ambiguity, so more reasons that might reinforce the behaviors of dependent teachers. It will be difficult for people with distrust and dependence to feel empowered; therefore, we propose the following hypothesis:

**Hypothesis 7 (H7):** *Distrust (a) and dependence (b) attenuate the positive association between distributed leadership and empowerment.*

Empowerment and organizational identification are important predictors for two of the most researched factors in relation to teacher work, namely work self-efficacy and work satisfaction. Teacher self-efficacy refers to a set of beliefs that teachers are able to effectively guide students towards their educational goal achievement, are able to overcome difficult work situations and are able to engage students in educational activities, helping them to overcome limitations and setbacks [44]. Work satisfaction, on the other hand, reflects positive attitudes towards work associated with positive emotional states resulting from evaluative cognitions in relation to work experiences [45]. Meta-analytic evidence supports the positive association between work self-efficacy and teacher performance [46] as well

as with commitment to the teaching profession [47]. Based on meta-analytic evidence and the fact that teacher self-efficacy at work is a significant predictor for students' academic achievement and their engagement with educational activities, we consider work self-efficacy an accurate proxy for teacher work performance. Meta-analytic evidence also shows that organizational identification benefits a broad range of work-related attitudes and behaviors [30,48]; therefore, we expect that identification is one of the mechanisms through which distributed leadership impacts teacher performance outcomes. Because empowerment and organizational identification impact work performance as well as various work-related attitudes, including work satisfaction [49–52], we see teacher empowerment and organizational identification as key mechanisms that explain the work-related outcomes of distributed leadership [30,53,54]. In line with the theoretical arguments presented above, we expect that the relationship between distributed leadership on the one hand and work self-efficacy and satisfaction on the other is mediated by empowerment and organizational identification. This mediation model can explain how distributed leadership can be a double-edged sword, as it is expected to increase empowerment as well as decrease identification.

**Hypothesis 8 (H8):** *Organizational identification and empowerment mediate the association between distributed leadership on the one hand and work satisfaction and self-efficacy on the other hand.*

## 5. Methods

*Sample and Procedure*

We distributed an online survey to a large sample of school teachers from different regions from Romania. The final sample consisted of 3528 teachers (350 of them were male) with an average age of 42.86 years old (SD = 9.58) nested in 329 schools from different regions in Romania. Our sample consisted mostly of female teachers, yet this gender imbalance is aligned with the representation in the Romanian educational system, accordingly to National Institute Statistics of Romania, showing that the vast majority of Romanian teachers are female [55] (The Educational System in Romania, 2020). All teachers were employed in the public system of state schools, also dominantly representative in the Romanian educational system. We used a cross-sectional study design based on an online survey that was distributed among different schools, and the teachers were asked to fill it in. Participation was voluntary with no incentives associated with filling in the survey, answers were collected anonymously and participants could withdraw from the online survey any time.

In this survey, we included several variables. As control variables, we asked participants to report their gender and age. Moreover, in order to evaluate the main variables included in the theoretical framework, we used the following scales:

*Distributed leadership* has been defined as the leadership responsibilities and functions that need to be fulfilled by different people with varied levels and types of expertise [10,16,35]. We evaluated this variable by asking participants how many individuals fulfilled leadership functions in their school, and it was coded as a dummy variable (one leader versus multiple leaders). Distributed leadership was recoded as a dummy variable for further analyses with 0—a single leader and 1—multiple leaders in the school.

*Organizational identification* is defined as the perceived oneness of an employee with an employing organization and the feeling that they belong to it [27]. We evaluated organizational identification using a single pictographic item of organizational identification introduced in Shamir and Kark [56], a valid measure of organizational identification [57], as participants were asked to rate the strength of their identification with the organization by choosing among circles overlapping to varying degrees, ranging from 1 assigned to two non-overlapping circles ("I don't identify at all with my organization") to 7 depicting two fully overlapping circles ("I identify myself totally with the organization").

*Empowerment* was defined as employees receiving more autonomy, self-leadership and control of the work environment from their leaders [33,34,58], and we evaluated empowerment with a behaviorally anchored single-item measure reported in Cremers and Curșeu [50]: "Think of how leadership functions are exercised in your school and select the value that best describes your situation: 1 = Is restrictive and directive (we receive directions, clear instructions that limit our freedom to act) to 7 = Is empowering (gives us the power and autonomy to decide how to do our work)".

*Work self-efficacy* of teachers was evaluated with a 12-item scale developed by Evers, Brouwers and Tomic [44] to evaluate the extent to which teachers believe they can successfully perform various aspects of their daily jobs (guiding collaborative groups, engaging students, use innovative educational practices). As the items refer to the most important domains of teaching effectiveness, we consider this scale to capture well the perceived work performance of the teachers involved in this study. Examples of items were "I am able to foster cooperation in a group when the pupils experience difficulties in this", "If a student experiences difficulties in doing a task, I am able to help him or her on the right course", "Even when skeptical colleagues comment on it, I am able to keep on putting my back into innovative projects", rated on a 1 to 5 Likert scale. The Cronbach's alpha for this scale was 0.92, indicating an excellent reliability of the scale.

*Job satisfaction* was evaluated using a single item ("How satisfied are you with your current job?") adapted from Nagy [59]. Answers were recorded on a 5-point Likert scale from "1 = very unsatisfied" to "5 = very satisfied".

*Distrust and dependence cognitive schema* were evaluated using a selection of items from the Schema Questionnaire [26,38]. The items were selected based on their factor loading from a study that adapted the questionnaire for the Romanian population [60] (Curseu et al., 2000). For example, one item used for distrust was "I feel like people will take advantage of me" and one for dependence was "I see myself like a dependent person in my everyday life". We assessed both variables using a 6-point Likert scale ("1 = Completely untrue about me" to "6 = It describes me perfectly"). For distrust, Cronbach's alpha was 0.70, while for dependence Cronbach's alpha was 0.56, showing rather low reliability of this scale. Given the low Cronbach's alpha for the dependence scale we used the omega procedure [61] based on a factor analysis to further investigate the scaling behavior of the items. All items of the dependency scale loaded significantly in a dominant factor (loadings varying from 0.51 to 0.58); therefore, we decided to use the scale for our further analyses.

## 6. Results

The means, standard deviations and correlations for the variables included in the study variables are presented in Table 1.

**Table 1.** Means, standard deviations and correlations.

| | Mean | SD | 1 | 2 | 3 | 4 | 5 | 6 | 7 | 8 |
|---|---|---|---|---|---|---|---|---|---|---|
| 1. Gender | 0.58 | 0.49 | 1 | | | | | | | |
| 2. Age | 42.86 | 9.58 | −0.041 * | 1 | | | | | | |
| 3. Distributed leadership | 0.67 | 0.47 | 0.023 | 0.156 ** | 1 | | | | | |
| 4. Distrust | 2.44 | 0.94 | 0.048 ** | 0.021 | 0.042 * | 1 | | | | |
| 5. Dependency | 1.79 | 0.72 | 0.023 | −0.109 ** | 0.009 | 0.408 ** | 1 | | | |
| 6. Organizational identification | 5.86 | 1.06 | −0.086 ** | 0.069 ** | −0.058 ** | −0.192 ** | −0.150 ** | 1 | | |
| 7. Empowerment | 5.13 | 1.16 | −0.077 ** | −0.067 ** | −0.085 ** | −0.190 ** | −0.065 ** | 0.411 ** | 1 | |
| 8. Work self efficacy | 4.34 | 0.52 | −0.044 ** | 0.022 | −0.074 ** | −0.125 ** | −0.283 ** | 0.402 ** | 0.249 ** | 1 |
| 9. Work satisfaction | 4.21 | 0.95 | −0.026 | −0.076 ** | −0.063 ** | −0.158 ** | −0.088 ** | 0.351 ** | 0.310 ** | 0.231 ** |

Note: gender is coded as a dummy variable with 0 = male and 1 = female; distributed leadership was coded as a dummy variable with 0 = a single leader and 1 = multiple leaders; *** $p < 0.001$; ** $p < 0.01$; * $p < 0.05$.

Because our data were nested, such as teachers being nested in schools that differed in their distributed leadership practices, we used Multilevel Modeling to test our hypotheses. In these analyses, we took into account gender, dependence and distrust cognitive schema

as Level 1 variables, as these are variables that correspond to the lowest level of analysis (individual teachers). Moreover, distributed leadership as well as all the cross-product terms for the interaction with the two types of dysfunctional cognitive schema were entered as Level 2 variables, as they correspond to the school level of analysis. The two mediators, namely empowerment and organizational identification, were considered at both levels of analysis. All variables were grand mean centered before the analyses and the hypotheses were tested using the MLmed macro, Beta 2 version for SPSS developed by Rockwood and Hayes [62] to test multilevel mediation and moderation models. The results of the multilevel analyses are presented in Table 2 separately for the within as well as between schools models.

**Table 2.** Results of the multilevel mediation analyses.

| Variables | Empowerment | | Identification | | Work Satisfaction | | Work Self-Efficacy | |
|---|---|---|---|---|---|---|---|---|
| | Level 1 | Level 2 | Level 1 | Level 2 | Level 1 | Level 2 | Level 1 | Level 2 |
| Constant | 0.20 ** (0.06) | | 0.15 ** (0.05) | | 4.26 *** (0.03) | | 4.37 *** (.02) | |
| Gender | −0.09 (0.06) | | −0.18 ** (0.06) | | 0.03 (0.05) | | 0.05 † (0.03) | |
| Distrust (DIS) | −0.13 *** (0.04) | | −0.12 *** (0.04) | | −0.06 † (0.03) | | 0.03 † (0.02) | |
| Dependence (DEP) | 0.05 (0.05) | | −0.04 (0.04) | | −0.002 (0.04) | | −0.16 *** (0.02) | |
| Distributed leadership (DL) | | −0.34 *** (0.09) | | −0.22 *** (0.06) | | −0.08 † (0.05) | | −0.04 (0.03) |
| DISXDL | | −0.12 ** (0.04) | | −0.04 (0.04) | | −0.01 (0.03) | | 0.01 (0.02) |
| DEPXDL | | −0.02 (0.06) | | −0.14 ** (0.05) | | −0.04 (0.04) | | −0.03 (0.02) |
| Empowerment | | | | | 0.16 *** (0.02) | 0.12 *** (0.04) | 0.05 *** (0.008) | 0.01(0.02) |
| Identification | | | | | 0.22 *** (0.02) | 0.29 *** (0.05) | 0.15 *** (0.009) | 0.23 *** (0.03) |
| N | 3522 | | 3522 | | 3522 | | 3522 | |
| -2LL | 29,951.55 | | 29,951.55 | | 29,951.55 | | 25,503.67 | |
| AIC | 29,963.56 | | 29,963.56 | | 29,963.56 | | 25,515.67 | |

Note: DIS: distrust cognitive schema; DEP: dependence cognitive schema; gender is coded as a dummy variable with 0 = male and 1 = female; unstandardized regression coefficients are shown with standard errors between parentheses; DL: distributed leadership coded as a dummy variable with 0 = a single leader and 1 = multiple leaders; Level 1: within schools, Level 2: between schools; *** $p < 0.001$; ** $p < 0.01$; * $p < 0.05$; † $p < 0.10$.

As illustrated in Table 2, at Level 2, distributed leadership has a negative and significant association with organizational identification (B = −0.22, SE = 0.09, $p < 0.001$) as well as with empowerment (B = −0.34, SE = 0.06, $p < 0.001$). This pattern of results is fully aligned with Hypothesis 1 but it is opposite to Hypothesis 2. Moreover, at Level 1, distrust had a negative association with organizational identification (B = −0.13, SE = 0.04, $p < 0.001$) and with empowerment (B = −0.13, SE = 0.04, $p < 0.001$), therefore Hypothesis 3 was fully supported. The association between dependence and organizational identification and empowerment was not significant, therefore Hypotheses 4 and 5 were not supported. Dependence has, however, a significant negative association with work self-efficacy. From the hypothesized interaction effects, only the interaction effect between dependence and distributed leadership was significant for organizational identification (B = −0.14, SE = 0.05, $p = 0.009$), while the interaction between distrust and distributed leadership was significant for empowerment (B = −0.12, SE = 0.04, $p = 0.007$). We can therefore conclude that only Hypotheses 6b and 7a were supported by the data. Hypotheses 6a and 7b did not receive empirical support. The significant interaction effects are presented in Figures 1 and 2.

All indirect effects specified in Hypothesis 8 were estimated at Level 2 (between schools), as distributed leadership was introduced as a Level 2 predictor. The indirect association between distributed leadership and work self-efficacy was only mediated by organizational identification (indirect effect −0.05, SE = 0.02, 95%CI −0.08;−0.02, $p = 0.002$) while the indirect effect via empowerment was not significant, as the confidence interval included zero (indirect effect −0.005, SE = 0.008, 95%CI −0.02;0.01, $p = 0.53$). The indirect association between distributed leadership and work satisfaction was fully mediated by organizational identification (indirect effect −0.06, SE = 0.02, 95%CI −0.10; −0.02, $p = 0.003$) and empowerment (indirect effect −0.04, SE = 0.02, 95%CI −0.08, −0.01, $p = 0.01$). This

relation reflects a full mediation, as the remaining main effect of distributed leadership on job satisfaction was negative, yet not significant (B = −0.08, SE = 0.04, 95%CI −0.17; 0.007, *p* = 0.07). This pattern of indirect effects provides only partial support for Hypothesis 7, as only organizational identification was a significant mediator for both work satisfaction and self-efficacy, a result that is aligned with meta-analytic evidence supporting the critical role of identification for job-related attitudes and outcomes [30,48]. Empowerment mediates the association between distributed leadership and work satisfaction but not work self-efficacy.

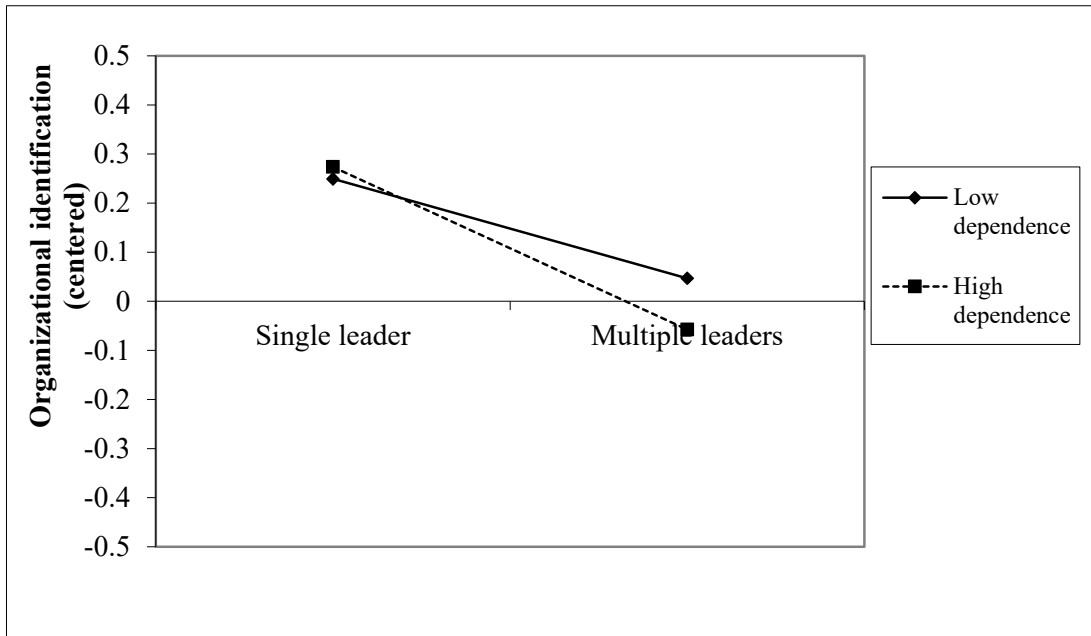

**Figure 1.** The interaction effect between distributed leadership and dependence on organizational identification (H6b).

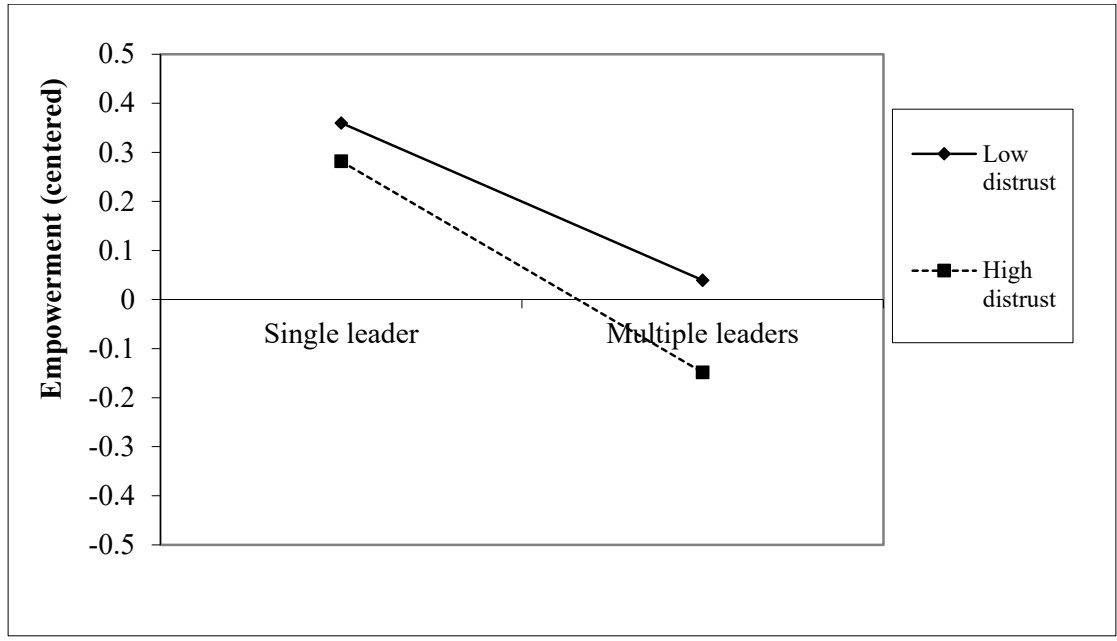

**Figure 2.** The interaction effect between distributed leadership and distrust on empowerment (H7a).

## 7. Discussion

The main aim of our paper was to test the association between distributed leadership and work satisfaction as well as work self-efficacy via two mechanisms that were expected to yield opposite indirect effects. Our results provide only partial support of our hypotheses. On the one hand, distributed leadership had a significant and negative association with both organizational identification and empowerment. Moreover, for the between school analyses, our results reveal an indirect negative association between distributed leadership on the one hand and work satisfaction and work self-efficacy on the other. Such results are surprising, as most empirical evidence to date has praised distributed leadership for its benefits for work and relational outcomes in schools. We believe our unexpected results support several explanations.

First, we estimated the effects of distributed leadership between schools by simply asking participants to report the number of leaders in their schools, and we did not use existing scales to evaluate all dimensions of distributed leadership. Most previous studies have used self-report scales that evaluated different dimensions of distributed leadership already pointing, in the phrasing of the items, towards the benefits of distributed leadership. For example, one of the most influential longitudinal studies [11] assessed distributed leadership dimensions such as the extent to which leadership makes decisions to enhance student achievements, encourage commitment and participation and allocate sufficient resources to support the achievement of educational goals [11]. In a similar vein, other scales frame the content of the items evaluating distributed leadership in line with their expected benefits like quality of support and supervision and effective communication cooperation among teachers [13.14] Therefore, there is no surprise that meta-analytic evidence, aggregating studies that used such scales, reports overall positive effects of distributed leadership in schools [9,63]. Other studies using distributed leadership scales asked participants to report the extent to which they are themselves engaged in performing leadership functions [2,4,19], and in such cases the positive association between distributed leadership and positive work and relational outcomes could be partially explained by Common Method Bias. Our results reveal significantly higher organizational identification and perceptions of leadership empowerment in school in which the leadership functions are exercised by a single person as compared to schools in which these leadership functions are distributed. We join the voices calling for more accurate and uniform definitions of distributed leadership in schools [7,9,16,35,63] in order to facilitate large-scale studies that allow the comparison of within-school differences in perceptions of distributed leadership (using self-report scales that evaluate the extent to which teachers themselves are have leadership roles) with the between-schools comparison (evaluating the number of formal leaders in each of the schools) of such differences.

Second, our study fully supports the predictions of the Social Identity Theory pointing towards the fact that on average, teachers in schools led by multiple individuals have difficulties in identifying themselves with their schools. Distributed leadership may generate lower distinctiveness of the school and as such decrease the strength of social identification with the organization. The significant moderation of dependence is also aligned with the optimal group distinctiveness explanation [28], as dysfunctional schema accentuate the negative association between distributed leadership and organizational identification. The surprising result concerns empowerment, such that schools led by multiple leaders actually fail to empower their teachers. A plausible explanation of such an effect lies in the insufficient training and planning of how leadership functions are to be exercised. Simply assigning someone a leader role does not necessarily mean that they can successfully fulfill leadership functions [16,35]. A direct implication of this interpretation is that when schools intend to distribute leadership functions across several teachers, they also have to make sure that sufficient training and resources are available to support these teachers to effectively exercise leadership. This second explanation is supported also by the fact that the negative impact of leadership on empowerment is accentuated by distrust. On average, in schools in which teachers are more inclined to be suspicious and perceive those

around them with greater doubt, the influence attempts of multiple leaders feel they are less empowered compared to schools in which, on average, teachers are less distrustful. School leaders need adequate training for how to deal with such suspicious interpersonal attitudes and behavioral tendencies, especially if such tendencies are generalized across teachers.

The third explanation that we put forward for the overall negative effects of distributed leadership is the cultural context in which we carried out our research [7]. Romania is a country scoring high in power distance, meaning that in general employees are ready to accept large power distance and find hierarchical organizational relations legitimate. In such a cultural context, distributed leadership may be perceived more negatively and lead to more negative outcomes, as it is not aligned with the expectation that the leading role is to be exercised by a single individual. On the one hand, we could imagine that a teacher that is exposed to several persons exercising leadership functions may feel they are actually constrained rather than empowered, given that interpersonal boundaries in multiple hierarchical relations restrict rather than empower. On the other hand, from the teacher in a leading position, the expectation is that her/his indications are simply accepted and followed and never challenged; therefore, the likelihood that they will actually empower other employees in a high-power-distance context is lower than in a low-power-distance one. To summarize, from the perspective of the target of social influence as well as from the perspective of the ones who exert social influence, distributed leadership has a low contextual fit with high-power-distance cultures.

Our results generally support the detrimental role of distrust cognitive schema on empowerment and identification, showing that teachers scoring high on distrust tend to feel less empowered and less identified with their schools as compared with teachers scoring low on distrust. The tendency to distrust others can have toxic relational consequences, as suspicious teachers may ultimately become socially isolated and feel deprived from any form of social support at work (instrumental and emotional) that is an essential resource for performing tasks effectively [52]. Our results did not support any of the hypothesized main effects for dysfunctional schema of dependence on identification and empowerment. The only significant association of dependence dysfunctional schema was with work self-efficacy, showing that teachers scoring high on dependence also tended to report lower levels of work self-efficacy. Such results can be explained through the association of dependence cognitive schema with general self-efficacy beliefs. Teachers who feel insecure and tend to seek advice and approval for their actions are also likely to have less confidence in their general skills and expertise; therefore, at work they tend to report lower work self-efficacy. These results open valuable avenues for future research on the role of dysfunctional cognition at work, as current research has focused almost exclusively on functional cognition [64], ignoring the workings of dysfunctional pattern of thought for interpersonal work relations and work outcomes.

Concerning the mediation claims, our results generally support what we hypothesized and show that organizational identification and empowerment are relevant mediators in the relation between distributed leadership on the one hand and work satisfaction and work self-efficacy on the other hand.

### 7.1. Limitations and Future Research Directions

Our study has several limitations. First, we used a particular operationalization of distributed leadership and looked at the extent to which leadership functions in schools are fulfilled by a single or multiple teachers. Future studies should use a combined approach and test in large samples, using multilevel modeling for the effects of perceptions of distributed leadership in combination with the effects of actual distribution of leadership functions across teachers [7]. Second, we used single items to evaluate satisfaction, organizational identification and empowerment. Although we have used established single-item measures, such measures provide only a global perspective on the concept being evaluated with no multidimensional estimation of such constructs (especially empowerment). Third, we collected data in a single cultural context and given the likely interplay of distributed

leadership with power distance, we join the voices calling for more cross-cultural investigations of distributed leadership and its implementation in schools [7,15]. Finally, our study was cross-sectional; therefore, the results are susceptible to common method bias. Nevertheless, we used interaction effects, and even in cross-sectional designs, such effects are not likely to be overestimated [65]; additionally, we used multilevel modeling, and most of the results concerning the implications of distributed leadership are actually based on aggregated school level scores and not the individual scores collected from the teachers.

### 7.2. Practical Implications

In our paper, we used the most basic way of operationalizing distributed leadership in schools, namely the extent to which leadership functions were assigned to one or several teachers. Based on this operationalization, the first practical implication derived from our study is that simply allocating leadership functions to different teachers does not yield positive outcomes in terms of organizational identification, empowerment and ultimately work efficacy and satisfaction. We urge school administrators to make sure that the teachers that take on such leadership functions are ready to cope with the relational complexity they entail. Ample resources and training opportunities should be made available in order to support teachers to "grow into the leadership roles". A second implication of our results refers to the critical role of dysfunctional cognitive schema. In particular schools, administrators should facilitate a work environment that does not prompt the activation of distrust cognitive schema, as such schema have detrimental influences on empowerment, identification and work satisfaction. Dependence cognitive schema reduce work self-efficacy and in combination with distributed leadership seem to significantly decrease organizational identification. We cannot clearly state whether the distributed leadership in schools creates a work environment that is conducive for the activation of distrust or dependence cognitive schema, yet we can state that especially when schools operate with distributed leadership systems, special attention should be devoted to creating a trusting and psychologically safe work environment for the teachers. Most certainly, such practical suggestions should be critically considered, as we derive such conclusions from a study conducted in a single cultural context and using a very specific operationalization of distributed leadership.

**Author Contributions:** Conceptualization: M.T., P.L.C. and A.F.M.; methodology, M.T., P.L.C. and A.F.M.; validation, A.F.M., P.L.C. and M.T.; formal analysis, P.L.C.; investigation, A.F.M. and M.T.; data curation, M.T., P.L.C. and A.F.M.; writing—original draft preparation, A.F.M., P.L.C. and M.T.; writing—review and editing, A.F.M., P.L.C. and M.T.; visualization, P.L.C.; supervision, P.L.C.; project administration, A.F.M., P.L.C. and M.T. All authors contributed equally to this paper. All authors have read and agreed to the published version of the manuscript.

**Funding:** This research was supported by the Wellbeing Institute, Cluj-Napoca Romania and the APC was funded by Babeș-Bolyai University Cluj-Napoca, Romania.

**Institutional Review Board Statement:** The study was conducted in accordance with the Declaration of Helsinki, and the protocol was approved by the the Scientific Council of the Babeș-Bolyai University Cluj-Napoca. The survey did not include questions with the potential to embarrass the participants or create distress, participation was voluntary and anonymous, and the participants could withdraw from the study at any time.

**Informed Consent Statement:** Informed consent was obtained from all participants involved in the study.

**Data Availability Statement:** The data analyzed in the current study are available from the corresponding author on motivated and reasonable request.

**Conflicts of Interest:** Authors have no conflict of interest to declare.

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
