# Peer review of "Does Distributed Leadership Deliver on Its Promises in Schools? Implications for Teachers’ Work Satisfaction and Self-Efficacy"

_education, doi:10.3390/educsci13101058_

Round 1

Reviewer 1 Report

I found this to be a very interesting study questioning the effectiveness of distributed leadership. The hypotheses are well-supported by the literature and the data collection, analysis, and conclusions are valid. I believe the content of the manuscript is ready for publication with minor revisions: The abstract mentions cognitive schema theory but the body of the paper focuses more on social identity theory and social interdependence theory. I also noticed that a few in-text citations are missing on page 6. Other than that, only copy editing is needed. Nice work!

For the most part, the manuscript is well written and conforms to standard conventions of the English language. However, there are several places throughout the manuscript where run on sentences, incomplete sentences, subject-verb agreement issues, etc. must be corrected prior to publication. I think the authors have done a good job to this point. I recommend a professional editor take over at this point.

Author Response

Answer: thank you for your appreciative and constructive remarks. We have added the missing citations on page 6. We have also proof read the manuscript, amended some of the sections and improved clarity in phrasing.

Reviewer 2 Report

The paper sets out to explore organisational identification and empowerment as mechanisms that may account for associations between distributed leadership and teacher-related outcomes, using cognitive  schema theory. Multi-level modelling analysis is used for a sample of 3528 teachers nested in 329 schools. Limitations of accepting a particular operational definition of distributed leadership and the extent to which leadership is a function of single or multiple teachers are acknowledged. Depending on the use of single items for evaluating satisfaction, organisational identification and empowerment is also highlighted as a limitation, as is use of a single cultural context. That said, the paper makes an important contribution to debate on distributed leadership in schools. Publication is recommended, subject to the inclusion of a formal of research question/s and careful proof-reading (e.g. line 60 – prototypically or prototypicality?; line 78 a or at?; and line 351, data is or are?).

Author Response

Answer: Thank you for your appreciative and constructive remarks. We have added the missing citations on page 6. We have also proof read the manuscript, amended some of the sections and improved clarity in phrasing. We have corrected the errors you mentioned in your review and also some other typos in other sections of the manuscript.

Reviewer 3 Report

This is a very interesting study, and your limitations are well stated. It would be important to do follow up on the items you mention. I am not sure I have seen such a long list of hypotheses in a single article, but given the methodology, type of data gathered, and the clear explication of limitations, I believe this is acceptable. 

The use of English is generally strong. The document does need a close read as there are a number of minor errors including missing letters, spacing, and errant/ unneeded words. 

Author Response

(The authors gave the same response as above.)

Reviewer 4 Report

A very well-written paper that consists a useful contribution to the field of education. Some minor editing needs to be carried out before finalization

A very well-written paper that consists a useful contribution to the field of education. Some minor editing needs to be carried out before finalization

Author Response

Answer: Thank you for your appreciative and constructive remarks. We have added the missing citations on page 6. We have also proof read the manuscript, amended some of the sections and improved clarity in phrasing. We have corrected some other typos in other sections of the manuscript. We have revised the discussion section and clarified the contributions of our paper to leadership in educational settings – we have also revised the introduction in order to clarify the distinction between the general distributed leadership literature and the empirical results reported in schools. Also we embed the discussion of our results in the education science literature and also in the conversations happening in the journal. We now refer to the following titles from ES to back up our contributions to the literature:

  1. Eryilmaz, N.; Sandoval-Hernandez, A. Is Distributed Leadership Universal? A Cross-Cultural, Comparative Approach across 40 Countries: An Alignment Optimisation Approach. Sci. 2023, 13, 218. https://doi.org/10.3390/educsci13020218
  2. Hickey, N.; Flaherty, A.; Mannix McNamara, P. Distributed Leadership in Irish Post-Primary Schools: Policy versus Practitioner Interpretations. Sci. 2023, 13, 388. https://doi.org/10.3390/educsci13040388
  3. Galdames-Calderón, M. Distributed Leadership: School Principals’ Practices to Promote Teachers’ Professional Development for School Improvement. Sci. 2023, 13, 715. https://doi.org/10.3390/educsci13070715
  4. Muntean, A.F.; Curșeu, P.L.; Tucaliuc, M. A Social Support and Resource Drain Exploration of the Bright and Dark Sides of Teachers’ Organizational Citizenship Behaviors. Sci. 2022, 12, 895. https://doi.org/10.3390/educsci12120895